# Genesis: Multimodal Driving Scene Generation with Spatio-Temporal and Cross-Modal Consistency

Xiangyu Guo[1*], Zhanqian Wu[2*], Kaixin Xiong[2*], Ziyang Xu[1], Lijun Zhou[2], Gangwei Xu[1],
Shaoqing Xu[2], Haiyang Sun[2 †], Bing Wang[2], Guang Chen[2],
Hangjun Ye[2], Wenyu Liu[1], Xinggang Wang[1 ✉]

[1]Huazhong University of Science and Technology    [2]Xiaomi EV
https://xiaomi-research.github.io/genesis/

## Abstract

We present Genesis, a unified world model for joint generation of multi-view driving videos and LiDAR sequences with spatio-temporal and cross-modal consistency. Genesis employs a two-stage architecture that integrates a DiT-based video diffusion model with 3D-VAE encoding, and a BEV-represented LiDAR generator with NeRF-based rendering and adaptive sampling. Both modalities are directly coupled through a shared condition input, enabling coherent evolution across visual and geometric domains. To guide the generation with structured semantics, we introduce DataCrafter, a captioning module built on vision-language models that provides scene-level and instance-level captions. Extensive experiments on the nuScenes benchmark demonstrate that Genesis achieves state-of-the-art performance across video and LiDAR metrics (FVD 16.95, FID 4.24, Chamfer 0.611), and benefits downstream tasks including segmentation and 3D detection, validating the semantic fidelity and practical utility of the synthetic data.

## 1 Introduction

As autonomous driving systems progress toward higher levels of intelligence, generating diverse, realistic driving scenarios [34, 30, 7, 44] has become essential for improving the robustness and safety of perception and planning modules. While recent advances have explored video generation and LiDAR sequence synthesis independently, achieving holistic multimodal consistency across visual and geometric modalities remains an open challenge, limiting the development of faithful world models that can simulate real-world sensor interactions.

As shown in Fig. 1, existing driving scene generation methods typically focus on generating data in a single modality, usually RGB videos [35, 10, 39, 50, 40, 42, 18, 26] or LiDAR point clouds [54, 53, 2, 3, 21, 22]. While these approaches have significantly advanced the field of driving scene generation, they overlook the synergistic potential of multimodal generation and lack consistency in aligning RGB videos with various sensor data, leading to limitations in real-world applications. Critically, they fall short of constituting a comprehensive world model, as they fail to capture the intricate dependencies between different sensory streams. Many of these methods rely on one-step layout-to-data pipelines conditioned solely on coarse spatial priors, such as BEV maps [36, 10, 43, 46, 39, 50, 40] or 3D boxes [12, 4, 33], which restricts their ability to capture complex scene dynamics and fine-grained semantics. To propel driving scene generation into the multimodal domain, UniScene [20] involves occupancy grids in for multimodal generation, while the acquisition of occupancy labels is

---

\* Equal contribution, the work was done when Xiangyu Guo was an intern of Xiaomi EV.
† Project lead (sunhaiyang1@xiaomi.com). ✉ Corresponding author (xgwang@hust.edu.cn).

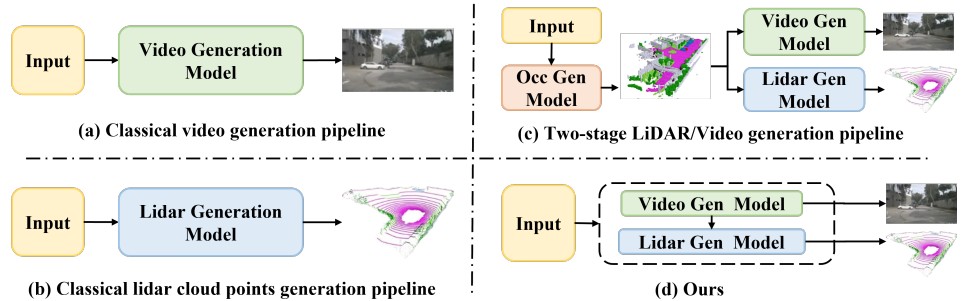

Figure 1: **Comparison of Multimodal Scene Generation Pipelines.** (a) *Video-only generation*, (b) *LiDAR-only generation*, (c) *Occupancy-guided dual-branch generation*, (d) **Our Genesis**.

highly costly, which severely limits its application. Moreover, most existing generation approaches generally rely on limited semantic annotations, typically in the form of coarse labels or generic captioning models, without fully leveraging the fine-grained descriptive capabilities of modern vision-language models (VLMs) [5, 23, 1]. This lack of structured semantic grounding restricts the fidelity, controllability, and contextual alignment of generated scenes, further distancing them from a truly grounded and actionable world model.

Therefore, advancing toward a realistic driving world model requires more than unimodal fidelity—it demands tight cross-modal alignment and fine-grained semantic grounding to ensure spatio-temporal coherence. However, existing methods suffer from three core limitations: (1) they often decouple video and LiDAR generation, weakening cross-modal consistency; (2) they rely on intermediate representations like occupancy grids that may cause information loss; (3) they provide limited semantic supervision, typically in the form of coarse layout maps or generic captions, which hinders scene-level controllability and realism.

To address these limitations, we propose **Genesis**, a unified joint generation framework tailored for autonomous driving that synthesizes multi-view RGB videos and LiDAR point clouds in a consistent and semantically grounded manner. Our framework introduces three key innovations:

- **Unified Multimodal Generation Architecture.** Genesis employs a unified pipeline where both video and LiDAR branches use shared conditional inputs, including scene descriptions and scene layouts, etc. To further guarantee the consistency between point clouds and image, BEV features extracted from projected image features are incorporated as conditional inputs into the Lidar diffusion model. This approach enhances cross-modality consistency without relying on intermediaries like occupancy grids.

- **Structured Semantic Captions via *DataCrafter*.** To improve semantic controllability, we introduce DataCrafter, a structured captioning module built upon vision-language models. It extracts multi-view, scene-level and instance-level descriptions that are fused into dense, language-guided priors. These captions provide detailed semantic guidance to both video and LiDAR generators, resulting in outputs that are not only realistic but also interpretable and controllable.

- **State-of-the-art performance.** Extensive experiments conducted on the nuScenes benchmark reveal that Genesis achieves the most advanced performance in both video and LiDAR metrics.

## 2   Related Work

**Driving Scene Video Generation.** Driving scene video generation has seen rapid progress, with many methods relying on structured spatial priors for controllability. BEVGen [35] conditions generation on BEV maps to encode road and vehicle layouts, but ignores height information, limiting its 3D representation capacity. BEVControl [46] addresses this by introducing a height-lifting module to partially restore scene geometry. MagicDrive [10] advances 3D-awareness through geometric constraints and cross-view attention, while MagicDriveDiT [9] leverages diffusion transformers for improved temporal fidelity. MagicDrive3D [8] further employs deformable Gaussian splatting for coarse 3D reconstruction. DriveDreamer [39] proposes hybrid Gaussians for temporally consistent synthesis of complex maneuvers. Voxel-based approaches such as WoVoGen [25] and Drive-WM [40]

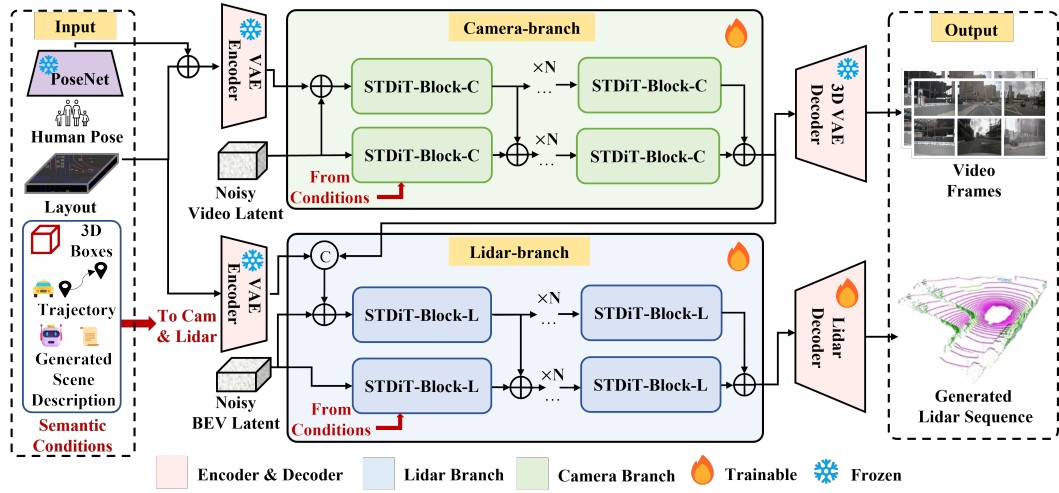

Figure 2: **Overview of the Genesis architecture for joint video and LiDAR generation.** A dual-branch design processes shared semantic conditions via camera and LiDAR pathways, using STDiT blocks for spatiotemporal generation and a BEV encoder for geometric alignment.

explore 4D voxel diffusion and latent substitution to model spatiotemporal dynamics. Despite progress, most methods remain unimodal and rely on weak or decoupled temporal priors, limiting their ability to generate globally consistent, semantically grounded sequences.

**LiDAR Point Cloud Generation.** LiDAR point clouds form the perceptual cornerstone of autonomous driving, enabling precise 3D understanding through sparse yet geometrically rich measurements. PointNet [31] and VoxelNet [52] laid the foundation for point- and voxel-based LiDAR processing. Building on these, neural radiance fields (NeRF) [28, 45, 41] incorporate LiDAR priors, improving training efficiency and geometric fidelity. Beyond static scenes, recent work emphasizes capturing spatiotemporal dynamics and generating high-fidelity, temporally consistent LiDAR sequences. Copilot4D [48] models long-range dependencies via LiDAR tokenization and hierarchical Transformers. ViDAR [47] employs video diffusion models to generate temporally consistent LiDAR sequences, while [53, 32] employ denoising and conditional diffusion to enhance geometry and consistency. While recent methods begin to capture temporal dynamics, they typically remain unimodal and underexplore semantic integration. In contrast, our framework jointly generates semantically grounded, spatiotemporally coherent sequences across both video and LiDAR modalities.

# 3 Method

In this section, we present Genesis, a unified generation framework designed to jointly synthesize multi-view video and LiDAR point cloud data with fine-grained semantic consistency.

**Overview.** As shown in Fig. 2, **Genesis** is architected as a unified world model with a dual-branch design that jointly synthesizes multimodal driving scenes, including videos and LiDAR point clouds. The camera branch leverages a DiT-based spatiotemporal diffusion backbone with a 3D-VAE encoder for fine-grained visual dynamics (Sec. 3.2), while the LiDAR branch employs a BEV-aware autoencoder with NeRF-style rendering and adaptive sampling for accurate geometric reconstruction (Sec. 3.3). Both branches are jointly conditioned on scene layouts, intermediate video latent features, and structured scene semantics to maintain strong cross-modal consistency. Semantic priors such as captions are derived from our DataCrafter module (Sec. 3.1) and injected as global conditioning to enhance alignment across modalities. More architectural details are provided in the supplementary material (Sec. A).

## 3.1 DataCrafter Module for Scene-Level Semantics

We propose **DataCrafter**, a structured captioning framework designed for multi-view autonomous driving videos. As illustrated in Fig. 3, given multi-view input $V = \{V_1, \ldots, V_K\}$, we segment it

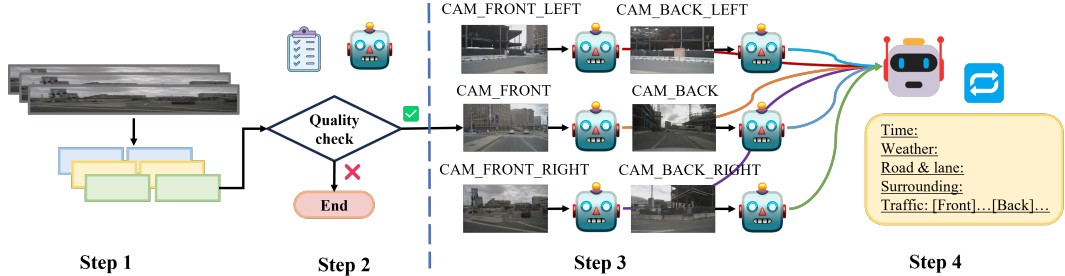

Figure 3: **DataCrafter pipeline for structured multi-view captioning.** Videos are segmented and filtered via a VLM-based quality checker (1–2), then per-view captions are generated and fused into coherent, structured descriptions (3–4). Steps 1–4 are used during training, while only 3–4 are used at inference.

into clips $\mathcal{C} = \{c_1, \ldots, c_N\}$ via a scene boundary detector. Each clip is scored by a visual-language model (VLM)-based module:

$$S(c_i) = \lambda_1 Q_{\text{clarity}}(c_i) + \lambda_2 Q_{\text{structure}}(c_i) + \lambda_3 Q_{\text{aesthetics}}(c_i), \tag{1}$$

where $Q$ terms represent VLM-derived subscores and $\lambda_i$ are fixed weights.

To ensure consistency across overlapping views, we introduce a *multi-view consistency module*. Let $\mathcal{Y}_k$ denote the caption from view $V_k$. Each caption is encoded via a pre-trained VLM:

$$\hat{\mathcal{Y}} = \mathcal{F}(\{\phi(\mathcal{Y}_k)\}_{k=1}^K), \tag{2}$$

where $\phi(\cdot)$ is a language encoder and $\mathcal{F}(\cdot)$ performs summarization and redundancy removal.

Finally, we produce a structured caption for each clip $c_i$:

$$\mathcal{Y}_i = \left\{ E_i, \{(o_j, b_j, d_j)\}_{j=1}^{M_i} \right\}, \tag{3}$$

where $E_i$ encodes global scene context (e.g., weather, road, time), and each object is represented by its category $o_j$, bounding box $b_j = (x_{1j}, y_{1j}, x_{2j}, y_{2j})$, and grounded description $d_j$.

This hierarchical design enables coherent and detailed scene-level and object-level captioning across time and views. More information about DataCrafter can be found in Sec. A.

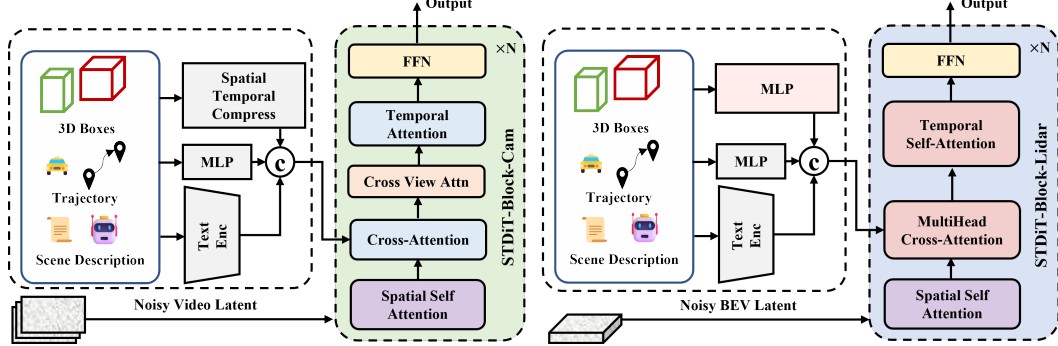

Figure 4: **Left:** STDiT-Block-C of Camera branch. **Right:** STDiT-Block-L of Lidar branch.

## 3.2 Video Generation Model

To ensure coherent multi-view video generation, a DiT-based diffusion backbone is extended with 3D-aware latent encoding and conditioned on scene-level priors—including road topology and linguistic descriptions via attention mechanisms, enabling spatial alignment, temporal consistency, and semantic fidelity.

**Structured Semantic Priors and Cross-View Conditioning.** To enable view-consistent and semantically grounded video generation, a structured BEV layout $\mathcal{S}^l = \{\mathcal{L}, \mathcal{H}, \mathcal{B}\}$ is constructed as

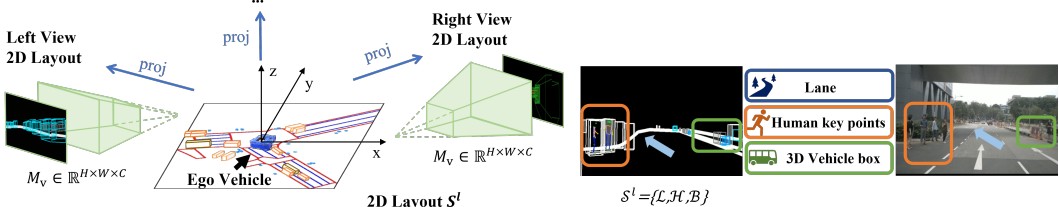

Figure 5: Illustration of the scene layouts projection pipeline, which enables view-consistent and semantically grounded video generation.

shown in Fig. 5, where $\mathcal{L}$, $\mathcal{H}$, and $\mathcal{B}$ denote lane segments, human pose keypoints, and 3D vehicle bounding boxes, respectively. Each element in $\mathcal{S}^l$ is projected onto the 2D image plane of view $v$ using calibrated intrinsics $K_v$ and extrinsics $E_v$, resulting in a set of view-specific semantic control maps $\mathcal{M}_v \in \mathbb{R}^{H \times W \times C}$. These projected maps—comprising layout curves, pose skeletons, and instance masks—are subsequently encoded and fed into a *Control-DiT* module.

Specifically, each block of the spatiotemporal diffusion transformer receives guidance from $\mathcal{M}_v$ via cross-attention, allowing the model to integrate structured priors at every denoising timestep. The forward process at timestep $t$ for view $v$ is formulated as:

$$z_v^{(t)} = \text{DiT}_\theta(z_v^{(t-1)}) + \text{CrossAttn}(z_v^{(t-1)}, \mathcal{M}_v), \tag{4}$$

where $\text{DiT}_\theta$ denotes the base diffusion transformer and CrossAttn represents the conditional modulation induced by $\mathcal{M}_v$.

**Pedestrian Pose Enhancement.** To enhance dynamic scene semantics, human keypoints are detected using YOLOv8x-Pose and projected into each view, forming semantic channels that augment layout and instance conditions.

**Latent Encoding via 3D VAE.** To ensure spatiotemporal consistency of control signals, a 3D Variational Autoencoder (VAE) is employed to compress the multi-frame scene layouts $\mathcal{S}^l$ and images into a latent representation $z \in \mathbb{R}^{f \times c \times h \times w}$. The decoder reconstructs the RGB images from the denoised latents.

**Scene-Level Caption Conditioning.** To encode high-level semantics, descriptive captions are generated via the *DataCrafter* module and processed using a pre-trained T5 encoder:

$$e_{\text{cap}} = \mathcal{E}_{\text{text}}(\text{caption}), \quad z_s = \mathcal{E}_{\text{layout}}(s). \tag{5}$$

Both embeddings are used as conditioning inputs via cross-attention in the DiT blocks:

$$\hat{z}_i = \text{CrossAttn}(q = z_i, k = [e_{\text{cap}}, z_s], v = [e_{\text{cap}}, z_s]). \tag{6}$$

**Spatiotemporal Control via STDiT-Block-Cam.** As shown in Fig. 4, our model integrates a Semantic-Aligned Control Transformer to modulate early-stage diffusion blocks. semantic features are injected via control attention:

$$h_i^{\text{ctrl}} = \text{Attn}(h_i^{\text{base}}, z_s), \quad i = 1, \ldots, K. \tag{7}$$

This module works in parallel with the core DiT pipeline, which includes spatial self-attention, cross-view attention, and temporal attention blocks to ensure view consistency and motion smoothness.

## 3.3  Lidar Generation Model

Lidar generation model is composed of two modules: a Point Cloud AutoEncoder Module for reconstructing point clouds and a SpatioTemporal Diffusion Module for generating Bird's-Eye-View(BEV) representation.

**Point Cloud AutoEncoder.** We design a BEV representation autoencoder to compress sparse and unordered point clouds into structured latent embedding. Inspired by [49, 48], we first voxelize the point clouds into BEV grids and then employ a Swin Transformer [24] backbone to downsample the BEV features by a factor of $8\times$ in spatial aspect and output the latent hidden feature with channel

4, which is further used in the SpatioTemporal Diffusion Module, as shown in Fig. 4. We decode point clouds from this latent embedding via a Swin-based decoder and a NeRF [28]-based rendering module. We adopt spatial skipping algorithm [48] to avoid the accumulated errors introduced by empty grids. The total training losses include vanilla depth L1 loss [15], occupancy loss [27], and surface regularization loss [17]. Furthermore, we propose a simple post-process algorithm to filter those noisy points generated from nerf rendering process. Specifically, we use filter point cloud that falls within grids, the occupancy value of which is smaller than the threshold.

**Spatiotemporal Diffusion with ControlNet-Branch.** With the latent embedding from the Point Cloud AutoEncoder Module, we design a diffusion network to learn this representation to generate realistic point clouds. Similar to the structure adopted in the video generation, we use a dual DiT-based network with ControlNet assistance. For semantic condition input, we use scene captions and scene layouts to provide structured scene priors. To keep cross modal consistency, we use the camera branch's RGB video outputs as another condition input. We adopt widely used LSS [29] algorithm to convert the perspective images to BEV feature. For geometric condition input, we encode 3D bounding boxes into embeddings for providing accurate foreground information. The image BEV feature is concatenated with the scene layouts latent feature and then send into the ControlNet branch as input. For a given latent token $z_i$ at block $i$, the cross-attention operation cloud be formulated as:

$$\hat{z}_i = \text{CrossAttn}(q = z_i, k = [e_{\text{cap}}, e_{\text{box}}], v = [e_{\text{cap}}, e_{\text{box}}]), \tag{8}$$

where the embeddings $e_{\text{cap}}$ and $e_{\text{box}}$ are derived from road sketches and 3D bounding boxes respectively. To ensure temporal coherence, the STDiT-Block-L applies multi-head self-attention operation. Given input $z'$, the tokens are updated as $\bar{z} = \text{MHSA}(z') + z'$. The entire LiDAR diffusion model is trained using a rectified flow schedule [6] to enhance generation quality.

## 4 Experiments and Main Results

### 4.1 Implementation details

Training and evaluation are conducted on the nuScenes [4] dataset, which includes 1,000 urban driving scenes (700 train / 150 val / 150 test) . Semantic occupancy labels at 2Hz are interpolated to 12Hz for dense supervision [10, 20]. Multimodal clips are sampled and evaluation follows the standard protocol [10, 20, 11] using 5,369 and 6,019 validation clips. Additional implementation details and training strategy are provided in the supplementary material (Sec. A).

### 4.2 Qualitative comparison and Versatile Generation Ability

High-quality video samples generated by our method demonstrate strong geometric fidelity and visual coherence across challenging scenes. As shown in Fig. 6 and 10, our model faithfully preserves vehicle shapes, lane structures, and environmental textures. In contrast, MagicDrive suffers from object deformation and layout artifacts, while Panacea exhibits hallucinated content and distorted backgrounds.

Beyond visual realism, the proposed framework enables controllable and diverse scene synthesis. By editing input captions, global scene attributes such as lighting and weather can be modified, as illustrated in Fig. 9. Furthermore, altering the ego-vehicle trajectory in the layout facilitates novel view generation, yielding coherent scene continuations with consistent geometry and appearance across perspectives, as shown in Fig. 11.

### 4.3 Video Generation Results

Tab. 1 reports the quantitative comparison of video generation performance under various generation modes. We evaluate our model in three configurations: without first-frame conditioning, with first-frame conditioning, and with noisy latent initialization. In all settings, our method consistently achieves superior performance across both FVD and FID metrics.

In the setting without first-frame conditioning, our method achieves an $\text{FVD}_{\text{multi}}$ of 83.10 and a $\text{FID}_{\text{multi}}$ of 14.90, outperforming prior works such as DriveDreamer-2 [50], MagicDrive-V2 [10], and Drive-WM [40]. With first-frame conditioning, our method further improves to 16.95 FVD and 4.24 FID, showing competitive results compared to MiLA [38] while maintaining temporal

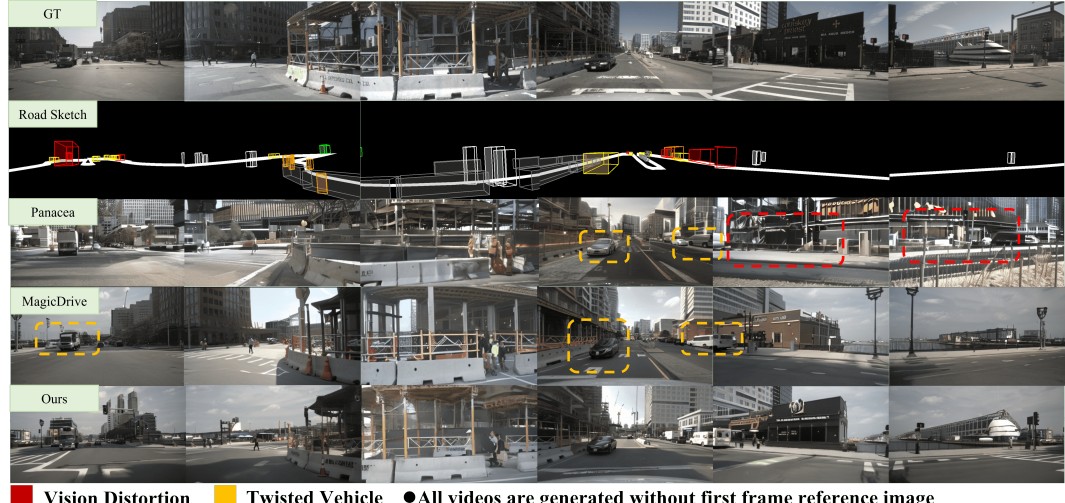

**■ Vision Distortion ■ Twisted Vehicle ●All videos are generated without first frame reference image**

Figure 6: **Qualitative comparison of video generation.** From top to bottom: (1) Ground-truth images, (2) Scene Layouts input, (3) Panacea [42], (4) MagicDrive [10], (5) Ours. Panacea suffers from hallucinated textures and geometric misalignment. MagicDrive shows vehicle distortion and broken structures. In contrast, **ours** preserves accurate layout, object shapes, and background integrity.

Table 1: Video Generation Comparison on nuScenes validation set, where green and blue represent the best and the second best values.

| Method | Gen. Mode | Multi-view | Video | Sample Num | Frame Num | $FVD_{multi}\downarrow$ | $FID_{multi}\downarrow$ |
|---|---|---|---|---|---|---|---|
| DriveDreamer-2 [50] | w/o first cond | ✓ | ✓ | – | – | 105.10 | 25.00 |
| MagicDrive-V2 [9] | w/o first cond | ✓ | ✓ | – | 16 | 94.84 | 20.91 |
| Drive-WM [40] | w/o first cond | ✓ | ✓ | – | – | 122.70 | 15.80 |
| Ours | w/o first cond | ✓ | ✓ | 5369 | 16 | 83.10 | 14.90 |
| MiLA [38] | w/ first cond | ✓ | ✓ | 5369 | 16 | 18.20 | 3.00 |
| DriveDreamer-2 [50] | w/ first cond | ✓ | ✓ | – | – | 55.70 | 11.20 |
| Ours | w/ first cond | ✓ | ✓ | 5369 | 16 | 16.95 | 4.24 |
| UniScene [20] | w/ noisy latent | ✓ | ✓ | 6019 | – | 70.52 | 6.12 |
| Ours | w/ noisy latent | ✓ | ✓ | 6019 | 16 | 67.87 | 6.45 |

consistency and structural fidelity. Under the noisy latent setting, we achieve 67.87 FVD and 6.45 FID on 6,019 samples, surpassing the previous best results reported by UniScene [20].

## 4.4 LiDAR Generation Results

Tab. 2 reports a quantitative comparison of LiDAR sequence generation performance between prior state-of-the-art methods and our proposed *Genesis* framework. Evaluation is conducted following the HERMES [51] protocol, using Chamfer Distance as the primary metric within a spatial volume of $[-51.2, 51.2]$ meters in the horizontal plane and $[-3, 5]$ meters in height.

Table 2: Lidar Generation Comparison on nuScenes validation set, where green and blue represent the best and the second best values. "gt_img" and "gen_img" indicate using ground-truth or generated images as BEV condition input, respectively.

| Method | Condition Type | Chamfer@1s $\downarrow$ | Chamfer@2s $\downarrow$ | Chamfer@3s $\downarrow$ |
|---|---|---|---|---|
| 4D-Occ [19] | gt_img | 1.13 | 1.53 | 2.11 |
| ViDAR [47] | gt_img | 1.12 | 1.38 | 1.73 |
| HERMES [51] | gt_img | 0.78 | 0.95 | 1.17 |
| Ours | gt_img | 0.611 | 0.625 | 0.633 |
| Ours | gen_img | 0.634 | 0.638 | 0.641 |

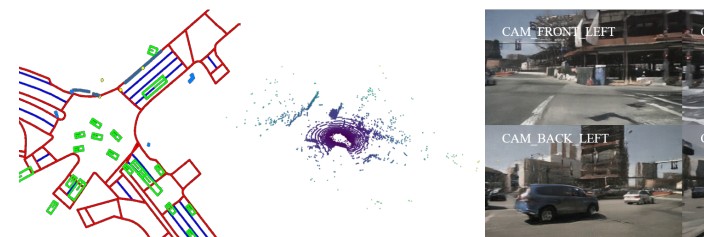

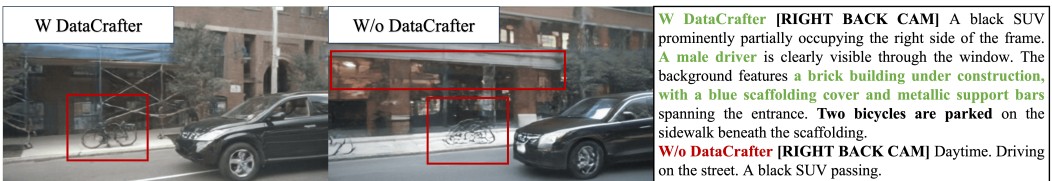

Figure 7: **Joint generation of LiDAR and multi-view video.** Our method generates spatially aligned LiDAR and camera views conditioned on a shared right turn T-junction layout.

| W DataCrafter | W/o DataCrafter | **W DataCrafter** [**RIGHT BACK CAM**] A black SUV prominently partially occupying the right side of the frame. **A male driver** is clearly visible through the window. The background features **a brick building under construction, with a blue scaffolding cover and metallic support bars** spanning the entrance. **Two bicycles are parked** on the sidewalk beneath the scaffolding. **W/o DataCrafter** [**RIGHT BACK CAM**] Daytime. Driving on the street. A black SUV passing. |

Figure 8: **Comparison with and without DataCrafter.** Our method yields semantically rich, structurally consistent scenes, while the baseline lacks fine-grained details and geometric accuracy.

**Genesis** consistently outperforms existing approaches across both short- and long-term horizons. At 1 second, it achieves a Chamfer Distance of 0.611, surpassing the previous best (0.78 by HERMES) by 21%. At 3 seconds, the advantage widens to a 45% relative reduction (from 1.17 to 0.633), underscoring the model's ability to maintain geometric fidelity over extended temporal predictions. Notably, the performance remains stable when replacing ground-truth images with generated images as conditional input, indicating strong robustness and effective cross-modal generalization. These results highlight the efficacy of our structured semantic conditioning and hierarchical generation design, establishing *Genesis* as a new state-of-the-art for LiDAR sequence synthesis, especially in long-horizon prediction scenarios where geometric consistency is critical.

### 4.5 Joint Generation Results

The proposed framework enables coherent joint generation of LiDAR and multi-view videos with consistent semantic and spatial alignment. As illustrated in Fig. 7, 12 and 13, our model maintains cross-modal correspondence even in complex scenes and over long temporal horizons (Fig. 14, 15). Dynamic objects and road structures are faithfully preserved across modalities, demonstrating the model's ability to synthesize structured scene dynamics with high temporal and geometric consistency.

### 4.6 Ablation Studies

We conduct an ablation study to assess the effectiveness of the proposed *DataCrafter* and *PoseNet* modules, which introduce structured semantic captions into the video generation pipeline. As shown in Tab. 3, removing *DataCrafter* causes a notable degradation in generation quality, with $FVD_{multi}$ increasing from 85.91 to 117.49 and $FID_{multi}$ rising from 15.20 to 22.32. Including the *PoseNet* module further improves performance, reducing $FVD_{multi}$ to 83.10 and $FID_{multi}$ to 14.90. These results underscore the importance of semantic guidance for maintaining temporal consistency and structural fidelity. As illustrated in Fig. 8, caption-conditioned generation produces sharper structures and more coherent layouts, whereas the ablated variant exhibits missing objects and visual distortions. In addition, *PoseNet* contributes fine-grained dynamic cues by localizing human anatomical keypoints.

Another ablation study on LiDAR generation is presented in Tab. 4 to assess the contributions of BEV latent features and first-frame conditioning. Removing first-frame input leads to a consistent performance drop across all prediction horizons (e.g., Chamfer@1s increases from 0.634 to 0.668), highlighting its importance for temporal guidance. Further excluding BEV latent features causes additional degradation, indicating their role in preserving spatial structure. These results demonstrate that both components are critical for achieving geometrically consistent long-range LiDAR synthesis.

Table 3: Ablation in the video generation model.

| Method | Gen. Mode | Sample Num | Frame Num | $FVD_{multi} \downarrow$ | $FID_{multi} \downarrow$ |
|---|---|---|---|---|---|
| baseline | w/o first cond | 5369 | 16 | 117.49 | 22.32 |
| w/ DataCrafter | w/o first cond | 5369 | 16 | 85.91 | 15.20 |
| w/ DataCrafter and PoseNet | w/o first cond | 5369 | 16 | 83.10 | 14.90 |

Table 4: Ablation in the Lidar generation model.

| Method | Chamfer@1s $\downarrow$ | Chamfer@2s $\downarrow$ | Chamfer@3s $\downarrow$ |
|---|---|---|---|
| w/o Img BEV Latent + w/o Ref. Frame | 0.661 | 0.669 | 0.673 |
| w/ Img BEV Latent + w/o Ref. Frame | 0.668 | 0.672 | 0.677 |
| w/ Img BEV Latent + w/ Ref. Frame | **0.634** | **0.638** | **0.641** |

## 4.7 Domain gap and Downstream task Utility

Our evaluation approach examines generated videos from two perspectives: the degree of reality measured by domain map metrics, and the functional utility demonstrated in downstream training applications such as 3D object detection tasks.

Table 5: **Domain gap on bev segmentation.**

| Method | mIoU↑ | mAP↑ |
|---|---|---|
| MagicDrive [10] | 18.34 | 11.86 |
| MagicDrive3D [8] | 18.27 | 12.05 |
| MagicDriveDiT [9] | 20.40 | 18.17 |
| DiVE [18] | 35.96 | 24.55 |
| Cogen [16] | 37.80 | 27.88 |
| Ours | **38.01** | **27.90** |

Table 6: **Effect of Multimodal Data Generation on 3D Object Detection.** Inputs of the methods in the table are both camera and lidar modalities.

| Method | mAP↑ | NDS↑ |
|---|---|---|
| Baseline | 66.87 | 69.65 |
| Ours(+cam_gen) | 67.09 (+0.22) | 70.12 (+0.47) |
| Ours(+lidar_gen) | 67.69 (+0.82) | 70.58 (+0.93) |
| Ours(+cam&lidar_gen) | 67.78 (+0.91) | 71.13 (+1.48) |

As shown in Table 5, we generate videos according to conditions from validation set to metric the domain gap following [9]. Our method achieves the best mIoU (38.01) and mAP (27.90) on bev segmentation, surpassing prior methods like DiVE [18] and Cogen [16], demonstrating strong semantic alignment and visual fidelity. As shown in Tab.6, we evaluate the effectiveness of our generative data on the BEVFusion[22] framework for 3D object detection. Our approach yields consistent improvements across all settings, increasing the mAP from 66.87 to 67.78 and NDS from 69.65 to 71.13. Notably, joint generation of both camera and LiDAR modalities achieves the highest gains (+0.91 mAP / +1.48 NDS), demonstrating the complementary benefits of multimodal generation. These results validate the utility of high-quality synthetic data in enhancing downstream perception tasks, especially in data-scarce or long-tail scenarios.

## 5 Conclusion

We propose **Genesis**, a unified world model for joint multi-view video and LiDAR point cloud generation in autonomous driving. Integrating structured semantic priors via the *DataCrafter* module and ensuring cross-modal alignment through a shared conditioning pipeline, it bridges visual and geometric modalities to enable high-fidelity, spatiotemporally coherent, and semantically consistent multimodal sequence synthesis. Experiments on nuScenes confirm Genesis achieves SOTA performance in both modalities, validating its robustness and scalability.

**Limitations and future work.** Despite Genesis's strong performance, several limitations remain. Training demands significant computational resources, motivating future efforts in model compression or efficient architectures. The framework is currently evaluated on nuScenes, and generalizing to other domains—such as robotics or aerial perception—requires further adaptation. Addressing these challenges is essential for broader deployment in safety-critical systems.

**Acknowledgement**: This work was partially supported by the National Natural Science Foundation of China (No. 62276108).

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

# A    Technical Appendices and Supplementary Material

## (a) DataCrafter Setup

To enable structured semantic supervision during training, the *DataCrafter* module was developed based on the Qwen2-VL 7B vision-language model. To enhance its understanding of driving scenes and its ability to assess sample quality, the base model was fine-tuned using Low-Rank Adaptation (LoRA) on two domain-specific datasets: *OmniDrive* and *CoDALM*. During inference, the fine-tuned model evaluated multi-view clips along three predefined dimensions: *visual quality*—including clarity and aesthetics—and *semantic descriptiveness*. Only samples that met a predefined quality threshold were retained for training, ensuring the exclusion of visually poor or semantically inconsistent sequences. The selected high-quality images were then forwarded to the captioning module. The captioning procedure is detailed in the following table.

You are provided with a set of descriptions of images captured by a vehicle's multiple cameras. Please carefully analyze these descriptions and answer the following questions. Your answer must strictly adhere to the format provided below, including all start and end tags for each field. All responses must be in English.

**Time:** Select one: "Daytime", "Night", "Indoor", "No visible sign"

**Weather:** Select one from "Sunny", "Cloudy", "Overcast", "Rain", "Snow" or "Night with no visible sign"

**Road Type:** Select one from "Highway", "Urban Road", "Rural Road", "Tunnel", "Bridge" or "No visible sign" **Road Surface:** Select one from "Asphalt ", "Concrete", "Gravel","resin (Indoor)" **Lane:** Select one from " "No visible sign","Single Lane", "Dual Lane", "Multi-Lane", "Other(should explain in Details)" and describe specifically what the lanes are.

**Environment      Type:**      Select      one      from      the      following      op-tions:"Highway","Roundabout","Intersection","Ramp","Tunnel","Parking Lot","Urban Road","Rural Road", "Bridge" or "Other(should explain in Details)"

**Surroundings Details:** Provide a general description of the environment (e.g., a busy street, a quiet neighborhood).

**Traffic:** Details: Describe the vehicles present in the scene. Include their spatial relationship to the ego vehicle—for example. Categorized by direction: Front, Left, Right, Behind. Each line follows the format: <position> <object> <action>.

## (b) Model Setup

The video generation pipeline is constructed upon a DiT-based architecture. The backbone is initialized with pretrained weights from MagicDrive [9], while the variational autoencoder (VAE) is initialized using the publicly available encoder from CogVideo-XL [14]. In contrast, the LiDAR generation branch is trained entirely from scratch. Both its diffusion backbone and 3D autoencoder are designed and optimized specifically for sparse geometry modeling, enabling independent learning of modality-specific representations without reliance on pretrained vision priors.

## (c) Training Strategy & Training Setup

A three-stage curriculum is adopted. (1) In the first stage, image-level generation is trained at 512×768 resolution to warm-start spatial representation learning. (2) The second stage focuses on video synthesis, where a two-phase training protocol is employed. Multi-resolution pretraining is first conducted by gradually increasing input resolution from 144p (144×256) to 900p (900×1600), paired with clip lengths ranging from 128 frames at lower resolutions to 6 frames at higher ones. This is followed by adapter-based fine-tuning at a fixed resolution of 360p (360×640) and 16 frames, where lightweight spatiotemporal adapters are inserted into DiT blocks. (3) The third stage performs joint training of video and LiDAR generation with shared conditioning signals, enabling cross-modal temporal alignment. For inference, video frames are generated at 900p with the first frame observed as input, consistent with prior work [8].

Training is performed with PyTorch using 64 NVIDIA H20 GPUs and mixed-precision acceleration. Stages 1, 2, and 3 are trained for 300, 800, and 200 epochs respectively. The optimizer is AdamW

with a weight decay of 0.01. A cosine annealing learning rate schedule is adopted with linear warm-up over the first 10% of steps. Learning rates are set to $2 \times 10^{-4}$ for the image and video stages, and $1 \times 10^{-5}$ for the joint stage. A global batch size of 1024 is used, distributed evenly across GPUs.

## (d) Evaluation Metrics

To comprehensively evaluate the quality of multimodal generation, we adopt a suite of metrics tailored to each modality. For video synthesis, we report Fréchet Inception Distance (FID) and Fréchet Video Distance (FVD), which assess spatial and spatiotemporal realism, respectively, in accordance with prior works [13, 37]. For LiDAR sequence generation, we employ Chamfer Distance at 1s, 2s, and 3s horizons, These metrics quantify both geometric fidelity and distributional alignment.

For evaluating downstream 3D perception utility, we adopt mean Average Precision (mAP) and NuScenes Detection Score (NDS) on generated sequences using a BEVFormer & BEVFusion [22] detector, following the protocol in [16]. These metrics provide a practical proxy for assessing the semantic controllability and structural consistency of generated data.

## (e) Supplementary visualization and quantitative analysis results

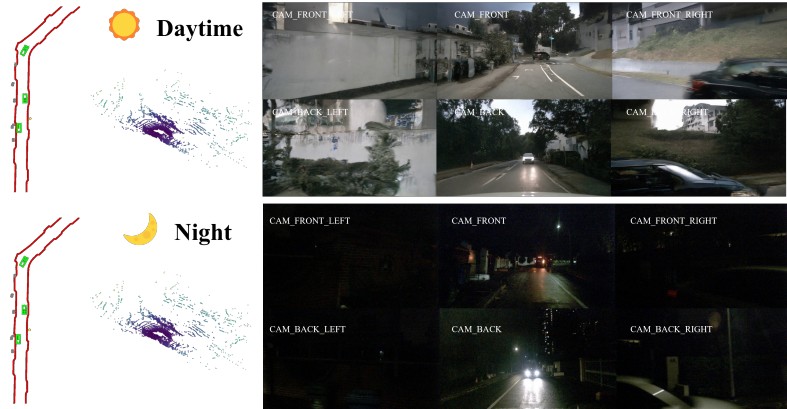

Figure 9: **Controllable generation across time-of-day.** By altering scene-level conditions, our method produces consistent multi-view videos aligned with the same underlying map and object layout, while adapting appearance to represent daytime and nighttime settings.

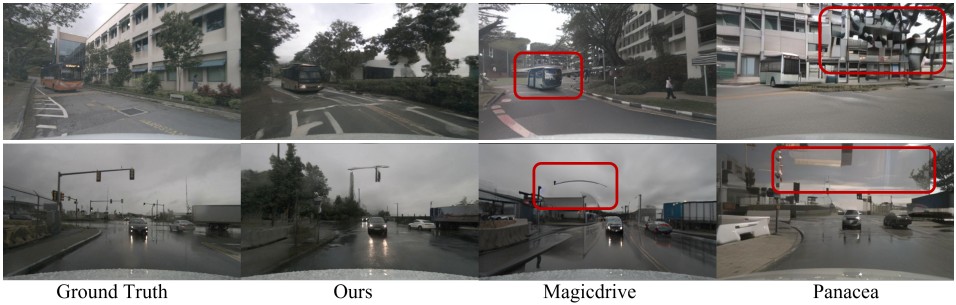

Figure 10: **Qualitative comparison of video generation quality.** Our method (second column) preserves accurate layout, object shapes, and background integrity. MagicDrive (third column) shows vehicle distortion and broken structures. Panacea (fourth column) often suffers from hallucinated textures and geometric misalignment.

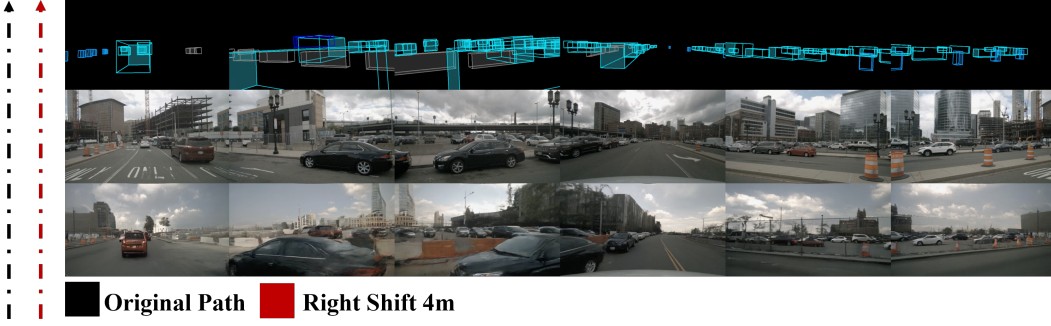

Figure 11: **Trajectory-conditioned novel view synthesis.** Given a ground-truth trajectory (middle), we modify the layout (top) by shifting the ego path 4 meters right (bottom). Our model generates plausible and consistent scenes across all views under these layout changes.

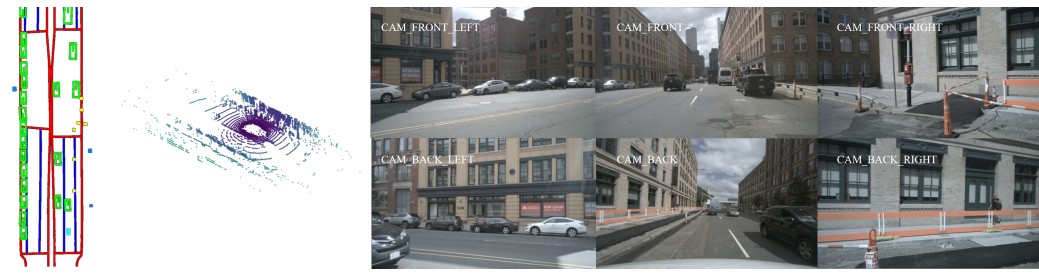

Figure 12: **Joint generation of LiDAR and multi-view video.** Our method generates spatially aligned LiDAR and camera views conditioned on a shared straight street layout.

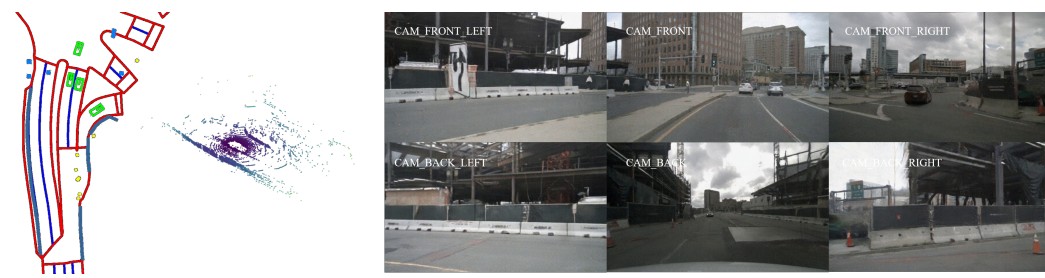

Figure 13: **Joint generation of LiDAR and multi-view video.** Our method generates spatially aligned LiDAR and camera views conditioned on a Busy junction layout.

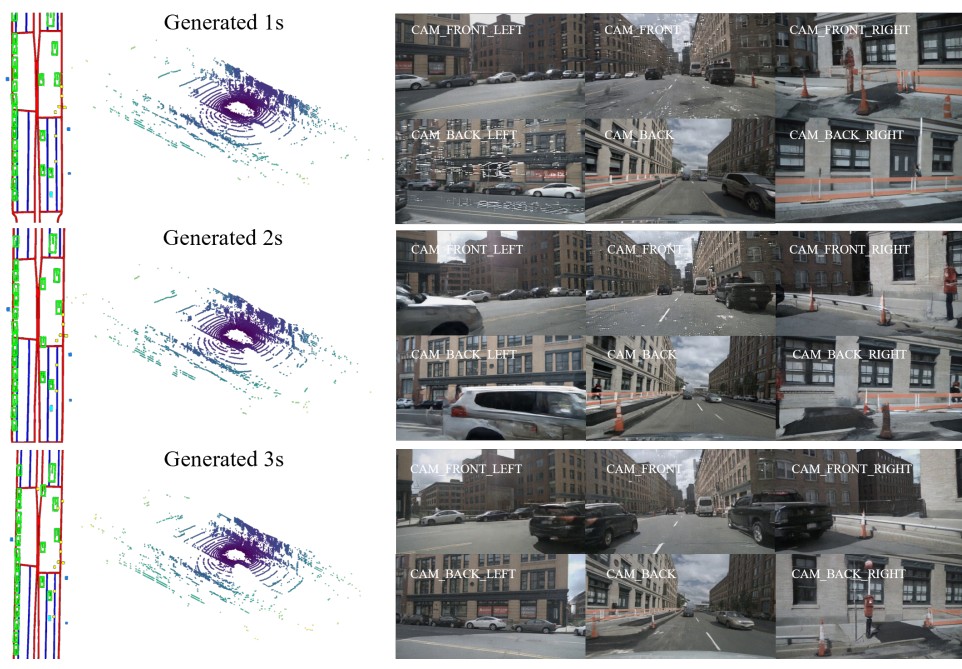

Figure 14: Long-term multi-view video generation over 3 seconds in an urban driving scene, conditioned on the straight street layout.

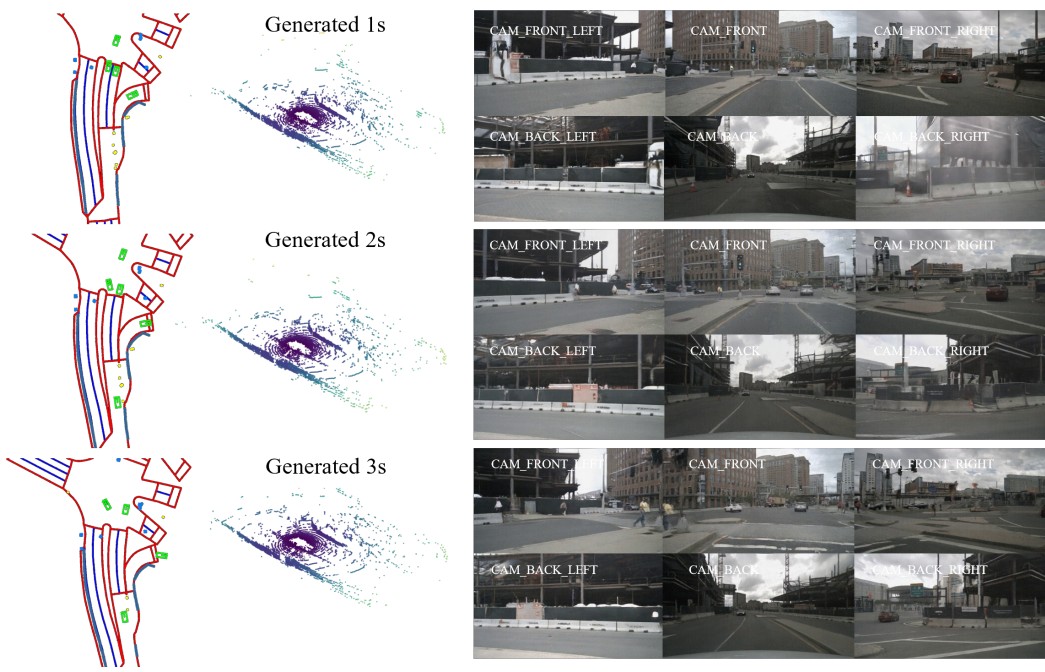

Figure 15: Long-term multi-view video generation over 3 seconds in an urban driving scene, conditioned on the busy junction layout.

