# OpenReview forum: "Genesis: Multimodal Driving Scene Generation with Spatio-Temporal and Cross-Modal Consistency"
_NeurIPS.cc/2025/Conference — NeurIPS 2025 poster_

### Official Review · Reviewer_EYgF · 2025-06-27

**Clarity:** 4
**Significance:** 3
**Originality:** 3
**Rating:** 5
**Confidence:** 4

**Summary:**

This paper introduces a multimodal driving scene generation framework named Genesis. It jointly generates multi-view driving videos and LiDAR with spatio-temporal and cross-modal consistency. It also introduced a VLM-based captioning module named DataCrafter to guide the generation with structured semantics. Extensive experiments on the nuScenes benchmark show that Genesis achieves state-of-the-art generation and alignment performance across video and LiDAR metrics

**Questions:**

No

**Ethical Concerns:**

["NO or VERY MINOR ethics concerns only"]

**Final Justification:**

The rebuttal has addressed my concern. I maintain my recommendation to accept the paper.

**Limitations:**

The limitations have already been discussed.

**Quality:**

3

**Strengths And Weaknesses:**

Strengths

1.	This paper proposes a unified framework for jointly generating videos and LiDAR point clouds, which has the potential to benefit multimodal downstream modules.

2.	The writing is clear and well-structured. Figure 1 effectively illustrates the comparison with previous pipelines.

3.	The generation quality is good for both videos and LiDARs.

Weaknesses

1.	While Genesis shows good generation quality, it remains unclear whether producing better replicas of training samples translates to meaningful improvements in downstream module performance. Since the ultimate goal of driving scene generation is to enhance downstream tasks, the paper would be stronger if it demonstrated how Genesis can generate novel and useful training samples.

2.	The inference cost is not discussed. It would be helpful to include a comparison of efficiency with other generation pipelines to assess the practicality of the proposed joint generation framework.

3.	The downstream evaluation is relatively limited. Expanding the evaluation to include additional tasks, such as lane detection, and testing on more benchmarks like Waymo, would further strengthen the paper.

---

> ### Author Rebuttal · Authors · 2025-07-29
>
> We sincerely thank you for the thoughtful review and constructive suggestions. Below we provide detailed responses to each concern raised. The additional experiments will be included in the revision.
> ### **Q1:  Downstream performance and capability of expanding the training samples**
>
> Thank you for this valuable suggestion. We have provided experimental results for 3D object detection task in the supplementary materials. We evaluate the effectiveness of our generated data on the BEVFusion framework for 3D object detection. Our approach yields consistent improvements across all settings, increasing the mAP from **66.87% to 67.78%** and NDS from **69.65% to 71.13%**. Notably, joint generation of both camera and LiDAR modalities achieves the highest gains **(+0.91% mAP / +1.48% NDS)**, demonstrating the complementary benefits of multi-modal generation. These results validate the utility of high-quality synthetic data in enhancing downstream  perception tasks, especially in data-scarce or long-tail scenarios.
>
> We primarily adopt the following two approaches for expanding training samples:
>
> 1. **Style transfer** was implemented based on existing scene layouts to enhance the model's adaptability to diverse meteorological conditions (e.g., changing from sunny to rainy weather);
>
> 2. **Editing and generalising for specific scene distributions.** This aspect is primarily validated on private dataset. For instance, if night-time cone detection metrics are low, we add various cones to the BEV layout and control the scene generation to be night-time; ultimately, the mAP metric improved from **16.47% to 17.26%** on the private dataset.
>
> ### **Q2:  Comparison of efficiency with other generation pipelines**
>
> We tested the inference cost on a single GPU NVIDIA H20 machine;
>
> 1. For the camera modality, generating a sequence of 17 frames with 6 views, each at a resolution of 424*800, takes 66.73 seconds. In contrast, MagicDrive-V2 requires a total of 231.18 seconds to complete the inference of 17 frames under the same setting. The main reason for faster inference speed is that our denoising step is 8, compared to 30 used in MagicDrive-V2.
>
> 2. For the LiDAR modality, generating 16 frames takes 57.72 seconds.  In contrast, UniScene first needs to generate semantic occupancy conditioned on BEV layouts; then, based on this, generate LiDAR point clouds. It takes a total of 63.2 seconds to generate 16 frames.
>
> We could draw a conclusion that Genesis achieves competitive inference efficiency in multi-modal generation.
>
>  ### **Q3:  Expanding the downstream evaluation**
>
> Thank you for your comment. In addition to 3D object detection tasks, we conduct domain gap testing on BEV segmentation tasks (including lane lines and foreground objects) on nuScenes dataset. Preliminary results indicate that our method achieves the best mIoU (38.01%) and mAP (27.90%) on BEV segmentation, surpassing prior methods such as DiVE and Cogen, demonstrating strong semantic alignment and visual fidelity.
>
> As for more experiments on  downstream evaluation, we conduct experiments on our private dataset. We trained Genesis on 100,000 clips (each clip lasting 30 seconds) and evaluate the downstream evaluation on 5000 clips. We focus on using generated data to solve specific issues. For example, we find that the mAP metric of  traffic cone detection during night-time is  low. Thus we add several traffic cones to the BEV layout and control the scene generation to be night-time. The quantity ratio of synthetic data and real data is 1:8.
>
> |Model|Detection category | Scenario| mAP(%) |
> |--|--|--|-|
> | Baseline| vehicles | night  | 69.71 |
> | Baseline + Synthetic Data| vehicles | night  | 71.36 (+1.65) |
> | Baseline| cone |night  |16.47 |
> | Baseline + Synthetic Data| cone |night | 17.26 (+0.79) |
>
> As shown in the above table, with the mixture training of real data and generated synthetic data, traffic cone's mAP metric improves from **16.47% to 17.26%** on our private dataset.

---

> > ### Comment · Reviewer_EYgF · 2025-08-05
> >
> > The rebuttal has addressed my concern. I maintain my recommendation to accept the paper.

---

> > > ### Author Response · Authors · 2025-08-05
> > >
> > > Dear Reviewer EYgF,
> > >
> > > Thanks for your positive feedback on our work!
> > >
> > > I'm glad to hear that our rebuttal has addressed your concerns. We sincerely appreciate your time and effort in providing such meticulous reviews and insightful comments.
> > >
> > > Best regards,
> > >
> > > Authors

---

### Official Review · Reviewer_3gho · 2025-07-01

**Clarity:** 3
**Significance:** 2
**Originality:** 2
**Rating:** 4
**Confidence:** 4

**Summary:**

The paper introduces Genesis, a framework for the joint generation of multi-view driving videos and LiDAR point clouds. It features a two-branch architecture where a video diffusion model (DiT-based with 3D-VAE encoding) and a LiDAR generator (with BEV-aware autoencoders and NeRF rendering) are coupled through a shared latent space. To improve semantic control and alignment, the authors propose DataCrafter, a captioning pipeline leveraging vision-language models to generate scene-level and instance-level descriptions. The model is evaluated on the nuScenes dataset and shows competitive results on standard metrics (FVD, FID, Chamfer).

**Questions:**

1.	Please clarify in what fundamental ways Genesis differs from UniScene and MagicDrive3D, which also support layout-conditioned multimodal generation. Are the performance gains mostly due to better tuning, or are there architectural contributions that make a structural difference?

2.	Has Genesis been tested (even qualitatively) on datasets beyond nuScenes, such as Waymo Open Dataset or KITTI? If not, what are the expected challenges for generalization?

3.	The ablation study shows modest improvements from DataCrafter, yet the module adds substantial complexity. Can you provide a controlled experiment comparing against simple captioning (e.g., T5 without VLM-based filtering)? This would help justify the added engineering overhead.

**Ethical Concerns:**

["NO or VERY MINOR ethics concerns only"]

**Final Justification:**

After reviewing the authors’ rebuttal, I recognize that several of my earlier concerns about novelty and contribution have been addressed. Although I still view the architectural innovations as incremental, the additional explanations, comparisons with related work, and supplementary experiments have clarified the value of the proposed framework. I now consider the reasons to accept to slightly outweigh the reasons to reject, and therefore I am updating my rating to borderline accept.

**Limitations:**

yes

**Quality:**

3

**Strengths And Weaknesses:**

Strengths:

1.	The paper offers a framework that couples video and LiDAR generation within a shared latent space.

2.	The proposed DataCrafter module uses structured scene descriptions for fine-grained control, which improves interpretability and controllability.

Weaknesses:

1.	The core novelty claimed—joint multimodal generation—has already been explored in prior works such as UniScene, BEVFusion, and MagicDrive3D, which also discuss layout-conditioned or multi-modal generation pipelines. Genesis does not go far beyond these methods in either formulation or outcomes. The use of a shared latent space and VLM-based semantic conditioning feels like a composition of existing techniques rather than a novel innovation.

2.	The model relies heavily on existing components (DiT, NeRF, ControlNet, BEV encoders, VLM captioners) and integrates them in a fairly straightforward way. While integration is non-trivial, there is little architectural or algorithmic novelty beyond known design patterns.

3.	The improvement margins are incremental compared to prior works like MagicDrive-V2 or UniScene. The broader impact is limited by the paper's tight focus on nuScenes only, with no discussion of transferability to other autonomous driving datasets or domains.

---

> ### Author Rebuttal · Authors · 2025-07-30
>
> We sincerely thank you for the thoughtful review and constructive suggestions. Below we provide detailed responses to each concern raised. The additional experiments will be included in the revision.
>
> ### **Q1: Explanation of model innovation**
>
> We would like to claim that our model is not a simple integration of existing design paradigms. Genesis is purposefully crafted to address the core challenge of **multi-modal data generation for autonomous driving scenarios**. We build our technical framework from the first principles: since images and point clouds are fundamentally different sensor modalities capturing the same physical scene, our approach begins by extracting core scene elements, including scene layout, foreground object bounding boxes, and semantic descriptions.
>
> These elements are then processed in a sensor-specific manner:
>
> 1. For the camera branch, the 3D conditions (layout and bounding boxes) are projected onto the image plane, yielding 2D layouts and 2.5D boxes (i.e., 2D boxes with depth) as conditional inputs.
> 2. For the LiDAR branch, the original 3D layout and bounding boxes are used directly as conditions.
>
> These processed representations serve as conditional inputs to a decoupled dual-DiT architecture, where we further introduce cross-modal consistency via image feature guidance.
>
> The key innovation of our framework lies in the shared condition guidance mechanism, which is the **first solution** capable of generating both image and point cloud video sequences without relying on the first-frame LiDAR input or explicit intermediate representations like occupancy grids.
>
> In addition, we implemented several detail-level enhancements:
>
> 1. During image generation, we project the layout into the image plane to reinforce viewpoint-consistency constraints;
>
> 2. For non-rigid object generation, we introduce keypoint-based control to improve geometric fidelity;
>
> 3. For the point cloud AutoEncoder, we propose a dedicated post-processing module designed to effectively filter out noisy points and enhance output quality.
>
> These designs reflect our commitment to build a principled, effective, and robust multi-modal generation system tailored for autonomous driving.
>
> ### **Q2：Explanation of the differences between Genesis, UniScene and MagicDrive3D, and the real reason for performance gains**
>
> **(1) The fundamental differences between Genesis and UniScene**:
>
> The fundamental distinction between Genesis and UniScene lies in the fact that Genesis **does not require any additional structural representations**. In contrast, UniScene relies on ground-truth OCC labels to supervise the training of a dedicated OCC generation network. However, obtaining such OCC annotations is costly and labor-intensive, limiting scalability in real-world scenarios. In comparison, Genesis achieves high-quality generation without relying on any external structural supervision, making it more practical and generalizable.
>
>  **(2) The fundamental differences between Genesis and MagicDrive3D**:
>
> 1. The essential difference between Genesis and MagicDrive3D is that Genesis is a fully multimodal generation framework, while MagicDrive3D **does not support LiDAR generation**. MagicDrive3D employs a monocular depth estimation network to produce pseudo-LiDAR point clouds from generated images. However, significant distributional discrepancies exist between these synthetic point clouds and authentic LiDAR data distributions.
>
> 2. For image video generation, Genesis uses projected 2D layouts as condition rather that 3D layouts used in MagicDrive3D. Compared to 3D Layouts, utilizing 2D Layouts ensures better multi-view consistency and facilitates view generalization, as evidenced by the successful extension from 6 views to 11 views.
>
> 3. For image video generation, Genesis proposes a pedestrian pose optimization strategy for enhanced pedestrian generation, which solves the challenging in non-rigid object generation in video synthesis to some extend.
>
> 4. Finally, compared with MagicDrive3D, we introduce DataCrafter to provide structured semantic supervision, yielding outputs that are realistic, interpretable, and controllable.
>
> **(3) The real reason for performance gains**:
>
> We maintain that the primary reason behind our performance gains is the shared latent space design, which enables tightly coupled multimodal modeling and significantly improves generation quality and consistency.
>
> For the camera modality, we introduce targeted enhancements for non-rigid objects such as pedestrians and adopt layout control via 3D-to-2D projection to ensure higher structural fidelity.
>
> For the LiDAR modality, we pioneer the BEV-Layout control mechanism, which allows for precise control of point cloud structures while maintaining cross-modal consistency.
>
> ### **Q3：Genesis testing experiments on private dataset**
>
> Thank you for this important reminder. In addition to our experiments on the nuScenes dataset, we also conducted extensive training and evaluation on a proprietary internal dataset. Below, we briefly introduce its characteristics: night-time scenes account for 32.14%, urban roads make up 69.43%, multi-vehicle complex road conditions cover 62.63%.
>
> We trained the Genesis model on 100,000 clips (each 30 seconds long) and evaluated its performance on 3D object detection using both Camera and LiDAR modalities. The evaluation results are shown below:
>
> |Modality| Precision(%) | Recall(%) | F1 Score(%)|
> |-|-|-|--|
> |Camera(Real)| 46.1| 92.6| 61.5|
> |Camera(Generated)| 45.3 |89.1|60.1|
> |Domain Gap| -0.8| -3.5| -1.4|
> |LiDAR(Real) |44.3| 89.1| 59.2|
> |LiDAR(Generated) |44.5| 86.8| 58.9|
> |Domain Gap| +0.2| -2.3| -0.3|
>
> We adopt a domain gap evaluation protocol, which quantifies the performance difference between using real and generated data. The goal is to evaluate the quality of the generated samples, guiding us to improve realism and utility by minimizing this gap.
> Our procedure is as follows:
>
> 1. Replace the real images or point clouds in the internal dataset with the generated ones.
>
> 2. Evaluate downstream 3D detection performance on the mixed dataset.
>
> 3. Compare the results against the performance on the original real dataset.
>
> As shown in table above, the F1 Score drops only 1.4% for the camera modality, and just 0.3% for the LiDAR modality, even without using condition frames. This demonstrates that our generated point clouds already exhibit high geometric fidelity, making them suitable for direct replacement of real LiDAR data in training and evaluation.
>
> For scene-specific editing and generalization, we also validate the effectiveness using this internal dataset. For example, in the case of low mAP for cones at night, we inject cones in diverse distances into the BEV layout under a nighttime condition constraint. This finally leads to the mAP improvement from **16.47% to 17.26%**.
>
> The detailed comparison is shown below:
>
> |Model|Detection category|Scenario| mAP(%) |
> |-|-|-|-|
> |Baseline| vehicles |night| 69.71 |
> |Baseline + Synthetic Data| vehicles |night| 71.36 (+1.65) |
> |Baseline| cone |night|16.47|
> |Baseline + Synthetic Data|cone|night| 17.26 (+0.79) |
>
> These results demonstrate that our synthetic data generation preserves high fidelity and simultaneously enhances downstream task performance.
>
> ### **Q4: Motivation and rationale for introducing the DataCrafter module**
>
> We appreciate your attention to the DataCrafter module, and would like to clarify its motivation and design rationale as follows:
>
> 1. Aligned with Mainstream Paradigms in Video Generation
>
> The current dominant paradigm for text-guided video generation follows the VLM → Caption → T5 → Embedding pipeline, as adopted by works such as VideoCrafter. In this context, the VLM component is not an added complexity, but rather a structural necessity for semantic control. Our proposed DataCrafter module does not introduce an additional VLM, but instead offers a high-quality refinement of this paradigm by applying caption filtering, re-ranking, and quality control on the VLM outputs, thereby generating more accurate and structurally sound captions.
>
> 2. Effectiveness Validated via Ablation Studies
>
> Ablation experiments have shown that DataCrafter significantly outperforms traditional captioning methods (e.g., BLIP) in terms of key metrics such as FID. These results demonstrate that DataCrafter enhances semantic quality and improves the final video generation performance. The comparative results will be explicitly highlighted in the final version.
>
> 3. Controllable Computational Cost
>
> Most VLM-caption pipelines are implemented with offline preprocessing: caption generation is performed during data preparation for training, and cached during inference. Only rare scenarios require real-time processing. Moreover, the captioning module accounts for a negligible fraction of inference time. In our deployment, DataCrafter (including filtering and augmentation) constitutes **only 2.44%** of total inference time, indicating high efficiency.
>
> 4. Supported by Real-World Applications in Autonomous Driving
>
> Several prior works in autonomous driving support the value of high-quality captions. For instance, DriveGenVLM uses VLM-generated captions to assess semantic alignment, and DriveDreamer-2 leverages LLMs to improve video-text consistency. These studies collectively demonstrate that high-quality language conditions are not only beneficial but essential for enhancing cross-modal alignment and structural accuracy. DataCrafter is designed as a refined and enhanced realization of this paradigm.
>
> In summary, DataCrafter is a lightweight and effective enhancement to the VLM → Caption → T5 pipeline. It introduces no additional model components, focuses purely on improving the quality of caption generation and selection, and has a well-controlled computational footprint. Backed by practical success in autonomous driving scenarios, we believe DataCrafter offers a justified and valuable contribution to video generation tasks.

---

> > ### Author Response · Authors · 2025-08-07
> >
> > Thank you for your detailed review and valuable feedback on our paper. We have noted that the discussion deadline is set for August 8th (AoE). We would appreciate it if you could kindly let us know if you have any further questions or suggestions, so that we can make any necessary adjustments and improvements in a timely manner. We truly value the opportunity to engage with you and look forward to refining our work further based on your insights. Should you have any additional questions or require further information, please do not hesitate to let us know. We are more than willing to continue the discussion and address any additional feedback you may have.

---

### Official Review · Reviewer_FmRN · 2025-07-05

**Clarity:** 3
**Significance:** 3
**Originality:** 2
**Rating:** 4
**Confidence:** 5

**Summary:**

The paper presents Genesis, a novel framework for joint generation of multi-view driving videos and LiDAR sequences with spatio-temporal and cross-modal consistency. The authors address a significant challenge in autonomous driving research by integrating structured semantic supervision and cross-modal alignment into a unified generation pipeline. The results demonstrate state-of-the-art performance in both video and LiDAR generation tasks, highlighting the framework's effectiveness in producing high-fidelity, semantically rich, and temporally coherent driving scenes. The work is well-executed, technically sound, and presents a substantial contribution to the field.

**Questions:**

1. In the FID_mult indicator in Table 1, this method is not as effective as UniScene. What is the reason? What are the next improvement measures?
2. As shown in Table 6, the accuracy of Lidar detection is lower than that of Camera detection when using the synthetic data of this method for incremental training. What is the reason? What are the next improvement measures?
3. For downstream utilization (incremental synthetic data for training), the accuracy is still declining. What are the bad cases we have seen so far? What problems need to be solved in using synthetic data?

**Ethical Concerns:**

["NO or VERY MINOR ethics concerns only"]

**Limitations:**

yes

**Paper Formatting Concerns:**

No obvious paper formatting concerns.

**Quality:**

3

**Strengths And Weaknesses:**

Strengths:
1. The text and pictures are clear and direct, and the algorithm framework is simple and effective;
2. A unified generation framework is designed to jointly synthesize multi-view video and LiDAR point cloud data with fine-grained semantic consistency;
3. The system design is complete, integrating video generation, Lidar generation, and semantic controllable data synthesis into a complete production line, effectively leveraging the strengths of each module;
4. According to the task, some detailed building block improvements are designed, such as 3D-aware latent encoding and conditioned on scene-level priors, which are effective and easy to implement;

Weaknesses:
1. There is a lack of refined processing for lidar generation, and the detailed design is more focused on the video generation branch;
2. The overall architecture innovation is average, mainly integrating some industry-verified effective building blocks, finely constructing training data, and making end-to-end training work;
3. The evaluation is mainly based on upstream generation quality evaluation, and there is a lack of detailed discussion on the incremental training evaluation of generated data for downstream detection or planning tasks;

---

> ### Author Rebuttal · Authors · 2025-07-29
>
> We sincerely thank you for the detailed and constructive feedback. We address each concern below and will incorporate the suggested clarifications and improvements in the final version.
> ### **Q1: Detailed design and refining process of LiDAR point cloud generation**
>
> Our design philosophy and refinement efforts focus on two components: the Point Cloud AutoEncoder and the LiDAR Generation Model.
>
> 1. In the Point Cloud AutoEncoder, directly applying the NeRF algorithm to LiDAR rendering results in multiple noisy points. This occurs because the α-blending learns the average distance across multiple occupied grids along the same ray. To address this, we design a post-processing module that utilizes the occupancy feature to suppress these noisy points.
>
> 2. For the LiDAR Diffusion model, previous methods rely heavily on the reference frame to predict the future point cloud sequence, which is comparatively a simple task.  In contrast, we focus on generating controllable point cloud sequences without reference frame input. For the foreground point cloud generation, we achieve precise control through BEV layout foreground boxes and category-aware box embeddings. For the background point cloud, we leverage road topology and scene descriptions, along with BEV latent representations extracted from images.
>
> We believe our LiDAR generation framework effectively simulates realistic point cloud distributions and the metrics on the nuScenes dataset reflect the effectiveness of our design.
>
> ### **Q2: Explanation of model innovation**
>
> We emphasize that our model is not a simple integration of existing design paradigms. Genesis is purposefully crafted to address the core challenge of **multi-modal data generation for autonomous driving scenarios**. We build our technical framework from the first principles: since images and point clouds are fundamentally different sensor modalities capturing the same physical scene, our approach begins by extracting core scene elements, including scene layout, foreground object bounding boxes, and semantic descriptions.
>
> These elements are then processed in a sensor-specific manner:
>
> 1. For the camera branch, the 3D conditions (layout and bounding boxes) are projected onto the image plane, yielding 2D layouts and 2.5D boxes (i.e., 2D boxes with depth) as conditional inputs.
>
> 2. For the LiDAR branch, the original 3D layout and bounding boxes are used directly as conditions.
>
> These processed representations serve as conditional inputs to a decoupled dual-DiT architecture, where we further introduce cross-modal consistency via image feature guidance.
>
> The key innovation of our framework lies in the shared condition guidance mechanism, which is the **first solution** capable of generating both image and point cloud video sequences without relying on the first-frame LiDAR input or explicit intermediate representations like occupancy grids.
>
> In addition, we implemented several detail-level enhancements:
>
> 1. During image generation, we project the layout into the image plane to reinforce viewpoint-consistency constraints;
>
> 2. For non-rigid object generation, we introduce keypoint-based control to improve geometric fidelity;
>
> 3. For the point cloud AutoEncoder, we design a post-processing module to filter out noisy points.
>
> These designs reflect our commitment to build a principled, effective, and robust multi-modal generation system tailored for autonomous driving.
>
> ### **Q3: Detailed discussion on downstream detection task**
>
> We apologize for the confusion between Table 6 in the main text and Table 1 in the supplementary materials. Table 6 in the main text presents the **domain gap metric** for BEV segmentation tasks, calculated by replacing ground truth images and LiDAR in the validation dataset. Preliminary results show our method achieves the best mIoU (38.01%) and mAP (27.90%) in BEV segmentation.
>
> We display the **downstream detection task** in the table 1 in the supplementary materials. Our approach yields consistent improvements across all settings, increasing the mAP from 66.87% to 67.78% and NDS from 69.65% to 71.13%. Notably, joint generation of both camera and LiDAR modalities achieves the highest gains (+0.91% mAP / +1.48% NDS), demonstrating the complementary benefits of multi-modal  generation. These results validate the utility of high-quality synthetic data in enhancing downstream perception tasks, especially in data-scarce or long-tail scenarios.
>
> ### **Q4: Reasons for our method being less effective than UniScene in the $FID_{multi}$ metric and future work**
>
> Thanks for this great suggestion. Regarding the slightly inferior performance of our method compared to UniScene on $FID_{multi}$ metric, we attribute the difference to two key distinctions in the conditional image selection strategies between our approach and UniScene:
>
> 1. UniScene utilizes all camera views from the 0-th frame, while we did not use.
>
> 2. UniScene incorporates 3D occupancy as auxiliary supervision, where the semantic-enriched depth information within volumetric grids delivers fine-grained geometric cues.
>
> In contrast, our current version does not incorporate such supervisions, primarily for the following reasons:
>
> 1. Maintain the lightweight nature of our method and avoid reliance on precomputed semantic/depth modules;
>
> 2. Focus on enhancing the model’s intrinsic spatio-temporal modeling capabilities, rather than leveraging external structural hints.
>
> Thus, from a technical perspective, our method achieves comparable performance under significantly fewer conditional inputs, which demonstrates its strong modeling capacity and potential for scalability.
>
> **Future Work and Improvements:**
>
> We plan to further enhance the boundary capabilities of multimodal generation while maintaining model generality and inference efficiency, with the following specific steps:
>
> 1. Few-shot sample generation capability. A key role of generative models is to provide downstream modules with abundant high-quality training data. Expanding the model's capability boundaries to generate samples rare in the training set is worth exploring—for instance, generating point clouds and images of scenarios like "a dog crossing the road" or "a fallen traffic cone".
>
> 2. Inference efficiency optimization. Model inference speed is a major bottleneck, mainly due to VAE resolution compression and denoising steps. While VAE compression rates for images are extensively studied, developing efficient compression methods for LiDAR latent spaces emerges as a critical research frontier. Additionally, exploring methods to reduce denoising steps is also worthwhile.
>
> ### **Q5: Reasons for lower LiDAR detection accuracy compared to camera in incremental training with synthetic data**
>
> Thank you for highlighting the potential ambiguity in our previous wording. Table 6 does not present incremental training results; instead, it evaluates generation quality by replacing real data with generated samples to measure their impact on downstream performance.
>
> Regarding why "LiDAR detection accuracy is lower than that of cameras", we attribute it to differences in LiDAR sparsity and data quantity. On the nuScenes dataset, LiDAR point clouds are relatively sparse (with lower beam density) and limited in volume (only 700 training clips), making it harder for the model to accurately learn the distribution of foreground point clouds. This results in a larger domain gap in LiDAR performance compared to image generation.
>
> To verify this, we conducted a comparison using our private dataset, collected with the Hesai AT128 LiDAR (with denser beams) and trained on 100,000 clips. The domain gap metrics are as follows:
>
> |Modality|Precision(%)|Recall(%)| F1 Score(%)|
> |-|-|-|-|
> |Camera(Real)| 46.1| 92.6| 61.5|
> |Camera(Generated) | 45.3 |89.1|60.1|
> |Domain Gap|-0.8| -3.5| -1.4|
> |LiDAR(Real)|44.3|89.1| 59.2|
> |LiDAR(Generated) |44.5| 86.8| 58.9|
> |Domain Gap| +0.2| -2.3| -0.3|
>
> As shown, in a higher-density LiDAR setup with more data, the domain gap of LiDAR becomes smaller than that of the image modality. This further supports our analysis that data quality and quantity, especially LiDAR beam density, play a crucial role in determining the effectiveness of generative modeling across modalities.
>
> ### **Q6: Bad cases of synthetic data in downstream training and problems to be solved**
>
> Thank you for this valuable suggestion. In relation to your question about "the decreasing accuracy when using additional synthetic data for training", we would like to clarify that the results shown in the supplementary table actually demonstrate an improvement. When utilizing both image and LiDAR modalities together, we observed a 1.48% increase in NDS compared to training without synthetic data, indicating a positive impact on downstream performance.
>
> As for your concern about bad cases, we have identified two primary issues during our internal evaluations:
>
> 1. Blurry generation of distant small objects. At distances beyond 150 meters, generated objects often become too indistinct to confidently determine whether they represent vehicles or other entities.
>
> 2. Occasional hallucinations. Despite employing classifier-free guidance to regulate foreground object generation, we observed rare instances where the model hallucinates vehicles in regions not conditioned by any bounding boxes.
>
> To address these challenges, we believe the following steps are essential:
>
> 1. Strengthening quality control for synthetic data: We are introducing a combination of VLM-based automatic filtering and manual verification to ensure that only high-quality samples are used for downstream tasks.
> 2. Ensuring the quality of training data for generative models: By refining the curation pipeline and enhancing the fidelity of the training dataset, we aim to minimize hallucination and improve the overall reliability of the generation model.

---

> > ### Comment · Reviewer_FmRN · 2025-08-05
> >
> > Thank you for the authors' efforts in providing detailed rebuttal.
> >
> > [Q1] Resolved.
> >
> > [Q2] The key innovation of the framework lies in the shared condition guidance mechanism, which is the first solution capable of generating both image and point cloud video sequences without relying on the first-frame LiDAR input or explicit intermediate representations like occupancy grids.
> >
> > [Q3] The experiments are sufficient.
> >
> > [Q4] Resolved.
> >
> > [Q5] Clarified.
> >
> > The rebuttal has addressed my concern. I would like to raise my rating to 'accept'.

---

> > > ### Author Response · Authors · 2025-08-05
> > >
> > > Dear Reviewer FmRN,
> > >
> > > Thank you for your positive feedback on our work!
> > >
> > > We are pleased to hear that our rebuttal has addressed your concerns. We sincerely appreciate your time and effort in providing such meticulous reviews and insightful comments.
> > >
> > > Best regards,
> > >
> > > Authors

---

### Note · Authors · 2025-08-14

We thank the ACs, SACs, PCs, and all reviewers for dedicating their time and expertise to our work. We value the detailed comments and insights offered by all three reviewers in the review. Reviewer FmRN highlighted the clarity of our presentation, the simplicity and effectiveness of the algorithmic framework, and the completeness of our unified system for multi-view video and LiDAR generation with fine-grained semantic consistency. Reviewer 3gho commended the coupling of video and LiDAR generation within a shared latent space, and emphasized the DataCrafter module’s structured-scene control for enhanced interpretability. Reviewer EYgF recognized the framework’s potential to benefit multimodal downstream tasks, the clear and well-organized writing, and the high generation quality for both modalities. We have carefully addressed all reviewers’ concerns and revised the manuscript accordingly, guided by their valuable and insightful feedback. We once again thank the ACs, SACs, PCs, and all reviewers for their constructive feedback and contributions.

---

### Decision · Program_Chairs · 2025-09-17

**Decision:**

Accept (poster)

**Comment:**

In their paper, the authors introduce Genesis, a unified framework designed to jointly generate multi-view driving videos and LiDAR sequences. The novelty of the method is maintaining consistency across different views, over time, and between the video and LiDAR data.

All the reviewers are positive about the contributions of the papers, including: (1) the novelty of the proposed Genesis framework; (2) the experiments are sufficient to support the claims about the performance of the proposed method; (3) the paper is written and presented clearly. After the rebuttal, most of the concerns of the reviewers were addressed, and all the reviewers are happy with the current stage of the paper.

In my opinion, the contributions and originality of the proposed methods are sufficient for acceptance at NeurIPS. Therefore, I recommend accepting it in its current form. I encourage the authors to address the reviewers’ suggestions and integrate their feedback into the camera-ready version of their paper.